# Profit shifting from Nigeria to Europe: The impact on human rights

Rachel Etter-Phoya[1,2]*, Stuart Murray[3], Stephen Hall[4,5], Michael Masiya[6,7], Bernadette O'Hare[1]

1 School of Medicine, The University of St Andrews, St Andrews, United Kingdom, 2 Tax Justice Network, Lilongwe, Malawi, 3 Bitwise Ltd, Dundee, United Kingdom, 4 Department of Economics, University of Leicester, Leicester, United Kingdom, 5 University of Pretoria, Pretoria, South Africa, 6 African Centre for Tax and Economic Studies (ACTES), Blantyre, Malawi, 7 Harvard University, Boston, Massachusetts, United States of America

* rmep1@st-andrews.ac.uk

## Abstract

The United Nations Universal Declaration of Human Rights states that everyone is entitled to economic and social rights essential to survive and thrive (Articles 25 and 26) and everyone is entitled to a social and international order in which their rights and freedom can be realised (Article 28). These rights must be ensured through national efforts and international cooperation (Article 22), but many millions of people worldwide do not access their rights, including the right to clean drinking water, safe sanitation, healthcare, and education. Government revenue from taxes plays a crucial role in ensuring these rights. However, globally, 10% of corporate tax revenue is lost because multinational corporations shift their profits from where they operate. This study examines the impact of profit shifting on tax revenue in Nigeria, focussing on access to economic and social rights and governance. It estimates the impact of revenue gains made on profits shifted from Nigeria to European tax havens, using data on profits shifted published by Wier and Zucman in 2022 and the Government Revenue and Development Estimations (GRADE) model for the estimations. The findings reveal that if the Nigerian government had additional revenue equivalent to tax losses, an additional 500,000 Nigerians would have their right to drink clean water and nearly 800,000 their right to use basic sanitation each day, 150,000 children would have their right to education, and 11 children would have their right to survive each day (amounting to 4,063 children each year). Increased revenue would also improve governance. In contrast, the gains European tax havens make as destinations for shifted profits in terms of rights are almost negligible, given that almost all Europeans have those economic and social rights discussed in this paper fulfilled. The tax reforms championed by the Organisation for Economic Co-operation and Development (OECD), including 27 European member nations, to tackle aggressive corporate tax avoidance and tax evasion—in short, tax abuse—fall short of ensuring a suitable international order for rights to be achieved. To remedy this, all European countries must support negotiations on international tax cooperation at the United Nations. This should include reforms on regulating multinational corporations, particularly through unitary taxation with formulary apportionment. In the short- and medium-term, interim measures to mitigate

**Data availability statement:** This study uses data from 'The Missing Profits of Nations'

(by Thomas Tørsløv, Ludvig Wier and Gabriel Zucman) for estimates of profits shifted inward and outward and for tax rates applied by tax havens on inward-shifted profits. Wier and Zucman's (2022) 1975-2019 updated estimates are available for download here https://missingprofits.world/wp-content/uploads/2022/11/WZ2022.xlsb.xlsx and visually presented online here https://missing-profits.world/. We have used the University of St Andrews and University of Leicester online modelling tool, the Government Revenue and Development Estimations (GRADE) model. We use version V3.12.2:2024/10/17 to translate revenue gains and losses into indicators of access to several rights, including impacts on governance. The GRADE model is available for use with further information online here https://medicine.st-andrews.ac.uk/grade/research/.

**Funding:** Funding from this work comes from the Scottish Funding Council International Science Partnership Fund 2023-24 (SFC / AN/21/2023) and Prof Sonia Buist Global Child Health Research Fund. BOH received both awards. These funders had no role in the study design, data collection and analysis, the decision to publish, and the preparation of the manuscript.

**Competing interests:** The authors have declared that no competing interests exist.

the harmful impacts of profit shifting are necessary. Countries must take steps to raise the global minimum corporate tax rate, introduce unilateral measures to tax multinational corporations, improve tax transparency and information sharing with lower-income countries, and strengthen anti-avoidance rules.

## Background

Economic and social rights (hereafter rights) are essential for individuals to survive and thrive. These rights include access to clean drinking water, safe sanitation, healthcare, and education. These birthrights of every individual are enshrined in the Universal Declaration of Human Rights (Articles 25 and 26) [1], adopted 76 years ago and codified in international treaties and instruments. The Declaration emphasises that "everyone is entitled to a social and international order in which the rights and freedoms set forth in this Declaration can be fully realised" (Article 28). This builds on the founding United Nations (UN) Charter of 1945 that promotes equal rights and the self-determination of people through international economic and social cooperation (Article 55) [2]. Subsequently, in 1966, the UN General Assembly adopted two international treaties considered foundational for international human rights law, which came into force a decade later: the International Covenant on Civil and Political Rights and the International Covenant on Economic, Social and Cultural Rights [3,4]. In the latter, economic and social rights, like the right to health, sanitation, and education, are rights the state must respect by providing for them through resources and services [5].

To fulfil their human rights obligations, countries need to raise public financial resources, or in other words, "without resources, there are no rights" [6]. Consequently, countries are responsible for ensuring these rights are realised both through domestic efforts and international cooperation (Article 22). They are also responsible for the extraterritorial impacts of their domestic policies and actions and the decisions made by international financial institutions of which they are members, such as on lending and credit [7]. By signing and ratifying international treaties, countries commit to align their constitutions, laws, policies, and resource allocation with their obligations [8].

In this paper, we consider whether everyone has their entitlement to their rights met and whether an international order exists where these rights can be realised, with a focus on the revenue required. To do this, we analyse the revenue gains and losses associated with corporate profit shifting, specifically looking at Nigeria's (because there is available data) lost tax revenue due to profits shifted to tax havens and how this impacts rights and governance. We compare these losses with the gains made by the European tax havens that attract the shifted profits. Finally, we briefly explore what could be done to progress towards creating an international order where everyone's rights and freedoms can be fully realised.

### Tax, human rights and development

Government revenue is the linchpin between government duties and people's rights [9]. Taxation is the primary source of government revenue in most countries [10]. Taxation and human rights intersect in several ways. First, revenue mobilisation and utilisation significantly influence a government's fiscal space to spend, including on essential public services required to fulfil rights, such as the right to health and education. Second, domestic and international tax policy affects the (re)distribution of resources between and within states and the profile and location of income and assets. Third, taxation can also incentivise specific behaviours and increase the costs of actions that harm society

or the environment, which infringe on rights. Lastly, fiscal policy plays a vital role in mediating the relationship between those who govern and those who are governed, where well-designed systems improve state accountability and effectiveness for realising rights for all [9,11].

Today, human rights and development are widely accepted as two sides of the same coin [12], with taxation considered integral to both. For instance, Global South governments and civil society strongly advocated for a human rights-centred approach in the UN Agenda 2030 for Sustainable Development, which created the Sustainable Development Goals (SDGs). The SDGs serve as the globally agreed blueprint for national and international development and cooperation, incorporating taxation, which was notably absent in the predecessor Millennium Development Goals. The SDGs explicitly aim to "to realise human rights for all" [13]. There is significant overlap between international human rights and the targets and indicators established for the 17 SDGs [14], which include the rights outlined by the International Covenant on Economic, Social and Cultural Rights. For instance, the right to education maps from Article 13 of the Covenant onto SDG 4, which aims to ensure inclusive and equitable quality education. However, tensions between development and rights may exist, particularly when individual SDG targets can be achieved without national governments respecting human rights [15], and because no explicit human rights goals have been included [9].

In lower- and middle-income countries, revenue generated from corporate taxes is especially vital. Corporate income tax often constitutes nearly one-fifth of total tax revenues in Africa (18.72% across 32 countries) and Asia and the Pacific (18.2% across 31 countries), and 15% in Latin America and the Caribbean. In contrast, Organisation for Economic Co-operation and Development (OECD) countries have higher tax-to-GDP ratios but are much less dependent on corporate income taxes, constituting 10.2% of total tax revenues [16]. However, all governments encounter challenges in collecting corporate income tax; issues like tax expenditures—such as tax incentives to reduce corporate income tax rates or the tax base [17]—and tax abuse [18] lead to lower revenues. The latter, the focus of this paper, is driven by cross-border aggressive tax avoidance by multinational companies, which results in a global loss of about 10% of corporate income tax revenue, and this is much higher in specific jurisdictions [19]. In response to the scale of profit shifting, SDG 16.4 represents a ground-breaking effort, establishing the first globally agreed target to reduce illicit financial flows, including corporate tax evasion and avoidance. Illicit financial flows, as defined by the United Nations in the SDG indicator 16.4, are "financial flows that are illicit in origin, transfer or use, that reflect an exchange of value and that cross country borders" and include tax-motivated illicit financial flows (such as aggressive tax avoidance or profit shifting by multinational enterprise groups) [20].

The same year the SDGs were adopted, UN member states gathered in Addis Ababa at the Financing for Development Conference to discuss how the goals would be financed. Domestic resource mobilisation and international tax system reform took centre stage. The outcome document—the Addis Ababa Agenda for Action—emphasises that each country must effectively mobilise and use public resources, support an enabling international economic environment, and cooperate on international tax to reduce aggressive tax avoidance [21]. It calls for "scaling up international tax cooperation" to be able to tackle illicit financial flows effectively. That tax and aggressive tax avoidance—or tax abuse—affects individuals and society, as well as the ability of governments to finance public goods and services, is well known. Yet "the effort to analyse the implications of global tax 'wrongs' for human rights and scrutinise the content of tax rules through the lens of human rights law" is a relatively new development [9] upon which we build with this study.

## Evidence of government revenue impacts on rights and governance

Globally, many people still experience deprivations of economic and social rights essential for their survival, such as the right to basic sanitation and water (see Table 1). Research shows that low government revenue per capita is linked to inadequate coverage of these rights. In high-income countries, where government revenue per capita is high, coverage of rights approaches nearly 100%. Empirical evidence indicates that increased government revenue leads to higher spending independent of governance, especially in lower-income settings [24]. This increased spending enhances access to public services and improves health, education and mortality outcomes, ensuring more citizens have their economic and social rights fulfilled [25,26].

Reeves and colleagues studied 89 lower-income countries using a cross-national panel model with fixed effects [24]. Their research demonstrates that an increase in tax revenue by $100 per capita is associated with a $10 rise in government health spending and skilled birth attendance, which contributes to reduced mortality rates. They also argue that the composition of government revenue is important. For instance, consumption taxes applied to goods and services, impose the same tax rate on all taxpayers regardless of income, which typically makes it harder for lower-income households to afford essential goods, hence these taxes are considered regressive. In contrast, progressive taxes on income, profits, and capital gains—which respond to the ability to pay and increase as the taxable amount increases—do not adversely affect child survival outcomes.

In addition to directly impacting human rights coverage, increases in government revenue also have an indirect effect by improving governance, with some empirical support [27,28]. The quality of governance determines a government's effectiveness in ensuring budget allocation and expenditure translate into quality public services [29,30]. Additional government revenue, combined with its effects on improving governance, enhances the quality of public services, which is the most effective way to reduce inequality [31,32].

However, government revenue and governance are not the only factors determining the outcomes of rights coverage. Individual and household health and education outcomes are determined by various elements, including historical, geospatial and socio-economic factors and inequities [33–36]. At the population level, though, evidence shows that increased revenue leads to greater government spending on public services, positively affecting health determinants and educational outcomes [37]. With empirical evidence, this article contributes to progressing the position that "[t]axation is critical to the realisation of human rights in both developed and developing countries" [9] and generating tax revenue is one of the key pathways for governments to meet their human rights obligations [11]. Thus, when one state's policies or practices impact government revenue in another state, they also impact human rights.

**Table 1. Average government revenue per capita and coverage of rights in 2022 [22,23].**

| | Average government revenue per capita in constant 2015 USD | Basic sanitation coverage (%) | Basic water coverage (%) | Children who survive until they are five (%) | Children in primary school (%) | Children in lower-secondary school (%) | Children in upper secondary school (%) |
|---|---|---|---|---|---|---|---|
| High-income countries | 14,595.32 | 96.44 | 99.22 | 99.37 | 97.05 | 96.1 | 91.25 |
| Upper-middle income countries | 2,004.80 | 88.76 | 96.06 | 98.31 | 93.77 | 91.45 | 80.79 |
| Lower-middle income countries | 419.55 | 65.67 | 83.03 | 96.27 | 89.08 | 83.41 | 66.22 |
| Low-income countries | 86.51 | 36.37 | 63.52 | 93.48 | 72.79 | 64.75 | 43.92 |

## Extraterritorial obligations for the human rights impacts of cross-border tax policy

Two international treaties are the foundation of international human rights law: the International Covenant on Civil and Political Rights and the International Covenant on Economic, Social and Cultural Rights (ICESCR) [3,4]. The UN General Assembly adopted these in 1966, building on the Universal Declaration of Human Rights. Together, the three are often called the International Bill of Human Rights. All European states have ratified these covenants, and they are reflected the European Union's (EU) founding treaty, where EU nations agreed to uphold non-EU citizens' rights, especially children's rights, in the wider world, and international duties are grounded in the Universal Declaration of Human Rights, including Articles 22 and 28 [38].

Curtailing international aggressive corporate tax avoidance is only possible through reforming domestic and international tax policy, incorporating the state's obligations to people in other countries and not just to their own people, which has been the conventional approach within international human rights law jurisprudence. Türkelli categorises extraterritorial responsibilities as follows: 1) responsibility for a state's own acts or omissions resulting in human rights violations, 2) violations from acts or omissions by international organisations to which states are members, and 3) acts or omissions by non-state actors that are headquartered in a state which leads to violations, like a company, as the state is obliged to regulate these [39]. There is growing recognition, particularly among UN treaty bodies and experts, national human rights institutions, and governments in the Global South, that states have extraterritorial obligations for the human rights impacts of tax policy across all three categories of extraterritorial responsibilities. Table 2 presents a timeline of key comments by UN rights bodies, including experts and committees, on the implications of cross-border aggressive corporate tax avoidance for human rights and extraterritorial obligations of states (category 1).

The Maastricht Principles, drafted at a conference of scientists and former and current UN experts at the University of Maastricht in 2011, have also been particularly defining, influencing subsequent UN human rights committees and experts in elaborating on the extraterritorial obligations of states. These principles sought to clarify the extraterritorial duties of states, drawing from international law [60]. They also affirm that states are responsible for non-state actors, including corporations, and must influence and regulate businesses to protect economic, social, and cultural rights and cooperate to ensure effective grievance mechanisms and remedies where human rights have been violated [61]. Further, the principles outline that extraterritorial obligations may arise when a state's action or omissions lead to human rights violations abroad.

Over the years, the Committee on Economic, Social and Cultural Rights has elaborated on extraterritorial responsibilities. It "has indicated that the ICESCR may have an effect beyond the borders of States Parties, meaning that states may be bound by their obligations under the treaty when acting extraterritorially" [62] especially because since the covenant was drafted changes in economic and financial transactions has affected economic, cultural and social rights. Under the ICESCR (Article 2(1)), states are expected to take collective action to ensure that economic, social and cultural rights are fulfilled outside of their national territories. For example, countries party to the ICESCR may have extraterritorial obligations for headquartered multinational corporations operating in other countries. In 2017, according to the Committee's interpretive statement of the Covenant (General Comment No. 24) on state obligations, countries "should also encourage business actors whose conduct they are in a position to influence to ensure that they do not undermine the efforts of the States in which they operate to fully realise the Covenant rights—for instance by resorting to tax evasion or

**Table 2. Timeline of select UN human rights experts and committees' comments on extraterritorial obligations of tax policy on human rights (adapted and updated from [40,41]).**

| Year | Country, where relevant | UN body or expert | Quotation or statement |
|---|---|---|---|
| 2014 | | The Special Rapporteur on extreme poverty and human rights, Magdalena Sepúlveda Carmona (A/HRC/26/28) [42] | "The most straightforward way in which government revenues can facilitate compliance with human rights obligations is by providing resources for public goods, such as education and health services – goods that are critical to realising human rights and that ultimately benefit the whole of society" (para 2). "A State that does not take strong measures to tackle tax abuse cannot be said to be devoting the maximum available resources to the realisation of economic, social and cultural rights" (para 60). "Tax abuse by corporations and high net-worth individuals forces Governments to raise revenue from other sources: often regressive taxes, the burden of which falls hardest on the poor. Therefore, if States do not tackle tax abuse, they are likely to be disproportionately benefiting wealthy individuals to the detriment of the most disadvantaged" (para 60). "States should therefore take concerted and coordinated measures against tax evasion globally as part of their domestic and extraterritorial human rights obligations and their duty to protect people from human rights violations by third parties, including business enterprises" (para 62). "With regard to international cooperation and extraterritorial impact, each State should refrain from any conduct that impairs the ability of another State to raise revenue as required by their human rights commitments, and cooperate in creating an international environment that enables all States to fulfil their human rights obligations" (para 80). |
| 2016 | Switzerland | UN Committee on the Elimination of Discrimination against Women (CEDAW/C/CHE/CO/4-5) [43] | "[T]he State party's financial secrecy policies and rules on corporate reporting and taxation have a potentially negative impact on the ability of other States, particularly those already short of revenue, to mobilise the maximum available resources for the fulfilment of women's rights" (para 40c). It also recommended that Switzerland "Undertake independent participatory and periodic impact assessments of the extraterritorial effects of its financial secrecy and corporate tax policies on women's rights and substantive equality, ensuring that such assessments are conducted impartially, with public disclosure of the methodology and findings" (para 41a). |
| 2017 | | UN Committee on Economic, Social and Cultural Rights (E/C.12/GC/24) [44] | In the Committee's General Comment No. 24 (2017) on State obligations under the International Covenant on Economic, Social and Cultural Rights in the context of business activities, it stated, "Consistent with article 28 of the Universal Declaration of Human Rights, 87 this obligation to fulfil requires States parties to contribute to creating an international environment that enables the fulfilment of the Covenant rights. To that end, States parties must take the necessary steps in their legislation and policies, including diplomatic and foreign relations measures, to promote and help create such an environment. States parties should also encourage business actors whose conduct they are in a position to influence to ensure that they do not undermine the efforts of the States in which they operate to fully realize the Covenant rights — for instance by resorting to tax evasion or tax avoidance strategies in the countries concerned. To combat abusive tax practices by transnational corporations, States should combat transfer pricing practices and deepen international tax cooperation, and explore the possibility to tax multinational groups of companies as single firms, with developed countries imposing a minimum corporate income tax rate during a period of transition. Lowering the rates of corporate tax solely with a view to attracting investors encourages a race to the bottom that ultimately undermines the ability of all States to mobilize resources domestically to realize Covenant rights. As such, this practice is inconsistent with the duties of the States parties to the Covenant. Providing excessive protection for bank secrecy and permissive rules on corporate tax may affect the ability of States where economic activities are taking place to meet their obligation to mobilize the maximum available resources for the implementation of economic, social and cultural rights" (para 37). |
| 2018 | Cyprus | UN Committee on the Elimination of Discrimination against Women (CEDAW/C/CYP/CO/8) [45] | "The Committee is concerned that: The State party's financial secrecy policies, legislation on corporate reporting and taxation practices might have an adverse impact on the ability of other States, in particular those already short of revenue, to mobilize maximum resources for the realization of women's rights" (para 42a). "The Committee recommends that the State party: Undertake independent, participatory and periodic impact assessments of the extraterritorial effects of its financial secrecy and corporate tax policies and its commercial activities on women's rights within the State party and on the ability of third States to mobilize maximum resources for the advancement of women's rights" (para 43a). |
| 2018 | New Zealand | UN Committee on the Elimination of Discrimination against Women (CEDAW/C/NZL/CO/8) [46] | "It [the Committee] is also concerned about the lack of measures taken by the State party to fulfil its extraterritorial obligations with regard to tax avoidance, tax abuse and exploitation of weak economies in developing countries, which further reduces the resources available in those countries to advance women's rights and gender equality" (para 37). "In line with the Committee's general recommendation No. 28, undertake independent, participatory and periodic impact assessments of the extraterritorial effects of its financial and corporate tax policies on women's rights and substantive equality, ensuring that the assessments are conducted impartially and that the methodology and findings are communicated to the public" (para 37d). |

*(Continued)*

**Table 2.** (Continued)

| Year | Country, where relevant | UN body or expert | Quotation or statement |
|---|---|---|---|
| 2018 | Luxembourg | UN Committee on the Elimination of Discrimination against Women (CEDAW/C/LUX/CO/6-7) [47] | In its concluding observations on extraterritorial obligations in particular, the Committee noted its concern that "the State party's financial secrecy policies, its corporate reporting and taxation practices and its incentives for companies registered in Luxembourg and operating abroad have a severe impact on the ability of other States, in particular those already short of revenue, to mobilize the maximum available resources for the realization of women's rights" (para 15). |
| 2018 | | The Independent Expert on the effects of foreign debt and other related international financial obligations of States on the full enjoyment of human rights, particularly economic, social and cultural rights, Juan Pablo Bohoslavsky (A/HRC/40/57) [48] | In 2018, the Independent Expert on the effects of foreign debt and other related international financial obligations of States published a set of principles recommending how states should design economic reform policies. These "Guiding principles on human rights impact assessments of economic reforms" include the regulation of the financial sector "to identify, prevent, manage and fairly allocate the human rights risks created by financial instability and illicit financial flows" and "international, binational or regional regulation is crucial for efficiency in combating evasion, avoidance, tax fraud and illicit financial flows" (Principle 11d, Policy Coherence). <br> Further, "States have an obligation to provide international assistance and cooperation in order to facilitate the full realization of all rights. As part of their obligations with regard to international cooperation and assistance, States have an obligation to respect and protect the enjoyment of human rights of people outside their borders. This involves avoiding conduct that would foreseeably impair the enjoyment of human rights by persons living beyond their borders, contributing to the creation of an international environment that enables the fulfilment of human rights, as well as conducting assessments of the extraterritorial impacts of laws, policies and practices" (Principle 13). |
| 2019 | United Kingdom of Great Britain and Northern Ireland | UN Committee on the Elimination of Discrimination against Women (CEDAW/C/GBR/CO/8) [49] | CEDAW recommended that the UK and Northern Ireland "continue to adopt measures to combat money-laundering and tax evasion, including by establishing public registers of companies and trusts in all of its overseas territories and Crown dependencies and undertaking independent, participatory and periodic impact assessments of the national and extraterritorial effects of its financial secrecy and corporate tax policies on the rights of women" (para 20). <br> It also recommended that the UK "revise its corporate, trust, financial and tax legislation, policies and practices, with a view to fully realising the enjoyment by women of their rights under the Convention, both nationally and abroad" (para 20). |
| 2020 | Ireland | UN Committee on the Rights of the Child (CRC/C/IRL/QPR/5-6) [50] | "Ensure that tax policies do not contribute to tax abuse by companies operating in other countries, leading to a negative impact on the availability of resources for the realisation of children's rights in those countries" (para 10c). |
| 2021 | | The Independent Expert on the effects of foreign debt and other related international financial obligations of States on the full enjoyment of all human rights, particularly economic, social and cultural rights, Attiya Waris (A/HRC/49/47) [51] | The current Independent Expert laid out her intention in the report "Taking stock and identifying priority areas: a vision for the future work of the mandate holder" to elaborate on the impact of illicit financial flows as an impediment to States' commitments to SDGs and in recognising the "main users" of illicit mechanisms were multinational corporations (para 62). |
| 2021 | | UN High Level Panel on International Financial Accountability, Transparency and Integrity (FACTI Panel) [52] | Established in 2020, the UN FACTI Panel was tasked with reviewing current challenges and trends related to financial accountability, transparency and integrity. The panel made 12 recommendations in its report, on the basis that "Illicit financial flows (IFFs) — from tax abuse, cross-border corruption, and transnational financial crime — drain resources from sustainable development. They worsen inequalities, fuel instability, undermine governance, and damage public trust. Ultimately, they contribute to States not being able to fulfil their human rights obligations" (p VII). <br> "Illicit transactions are found everywhere, but they have a much heavier impact on developing countries. They undermine public service delivery, productive investment, public trust, the integrity of institutions and the rule of law, within and across borders. The impacts are greater on women and girls. This drain on resources does more than financial damage. It erodes trust in both social contracts as well as international governance systems. Meanwhile, it increases inequalities within and between nations. The drain also undermines the ability of States to respect, protect and fulfil human rights.[…] Above all, these illicit financial flows will continue to divert crucial resources away from sustainable development, even during a world-shattering crisis, when countries need them most" (pp. 3-4). |
| 2022 | Netherlands | UN Committee on the Rights of the Child (CRC/C/NLD/CO/5-6) [53] | The Committee recommended the Netherlands to "Conduct independent and participatory impact assessments of its tax and financial policies to ensure that they do not contribute to tax abuse by national companies operating outside the State party that lead to a negative impact on the availability of resources for the realisation of children's rights in the countries in which they are operating" (para 9c). |

*(Continued)*

**Table 2.** (Continued)

| Year | Country, where relevant | UN body or expert | Quotation or statement |
|---|---|---|---|
| 2022 | Panama | UN Committee on the Elimination of Discrimination Against Women (CEDAW/C/PAN/CO/8) [46] | The Committee observed "The negative impact of the State party's financial secrecy policies, corporate reporting and tax policies on women's rights in its territory and on the ability of other States parties to mobilize maximum resources for the advancement of women's rights" (para 39b).<br>"The Committee recommends that the State party: […] Conduct assessments of the impact of the State party's financial secrecy policies, corporate reporting and tax policies on women's rights and substantive gender equality, in the State party and in other States parties; adopt measures, including legislation and oversight mechanisms, to ensure that its domestic financial and professional services industries are not involved in transnational tax avoidance arrangements, which curtail the State party's capacity to achieve substantive gender equality; and undertake necessary reforms concerning tax policies with the aim of addressing inequality and ensuring the economic empowerment of women" (para 40b). |
| 2022 | | The Independent Expert on the effects of foreign debt and other related international financial obligations of States on the full enjoyment of all human rights, particularly economic, social and cultural rights, Attiya Waris (A/77/169) [54] | In the Independent Expert's report "Towards a global fiscal architecture using a human rights lens", she notes that "States have an extraterritorial obligation to ensure that fiscal law and policy respect and protect the human rights of people beyond their borders and to contribute to the creation of an enabling international environment and refrain from exerting undue influence on other States in ways that undermine their ability to fulfil their human rights obligations". (para 24) She recommends "(a) Reform the global taxation system as part of genuine efforts to combat illicit financial flows, in line with human rights law and standards, including extraterritorial obligations. At a time of multiple crises and weakened fiscal capacities at the domestic level, better international cooperation and assistance in the regulation, repatriation and taxation of flows from developing countries can assist in rebalancing the fiscal space. International cooperation and assistance are essential" (para 52a). |
| 2022 | Switzerland | UN Committee on the Elimination of Discrimination Against Women (CEDAW/C/CHE/CO/6) [49] | "The Committee notes with concern that the State party's tax and financial secrecy policies may have a negative effect on the ability of other States, particularly in the global South, to mobilize the maximum available resources for the implementation of women's rights" (para 21).<br>"The Committee reiterates its previous concluding observations (CEDAW/C/CHE/CO/4–5, para. 41 (a)) and recommends that the State party undertake independent, participatory and periodic impact assessments of the extraterritorial effects of its financial secrecy and corporate tax policies on women's rights and substantive equality, ensuring that such assessments are conducted impartially, with public disclosure of the methodology and findings" (para 22). |
| 2023 | Ireland | UN Committee on the Rights of the Child (CRC/C/IRL/CO/5-6) [55] | The Committee recommended Ireland "Ensure that tax policies do not contribute to tax abuse by companies registered in the State party but operating in other countries, leading to a negative impact on the availability of resources for the realization of children's rights in those countries" (para 13f). |
| 2023 | Ireland | UN Committee on Economic, Cultural and Social Rights (E/C.12/IRL/CO/4) [56] | "The Committee is concerned about the growing income disparities in the State party. It is also concerned about certain aspects of the State party's fiscal policy, including the low ratio of tax revenues to gross domestic product and the fact that certain transfers do not reach the population segments that they were intended to benefit. While acknowledging efforts to address tax evasion and cross-border tax abuse, the Committee is also concerned about reports that financial secrecy legislation and permissive corporate tax rules continue to hinder the ability of the State party, as well as other States, to meet their obligation to mobilize the maximum available resources for the implementation of the rights enshrined in the Covenant. The Committee is further concerned about the persistently low budget level for the realization of economic, social and cultural rights. The Committee regrets to note that it has received no information regarding the implementation of the Covenant in relation to negotiations with international organizations on fiscal consolidation policy and public debt (art. 2 (1))" (para 14).<br>The Committee recommended that Ireland, "(c) Strengthen measures to combat illicit flows, cross-border tax evasion and tax fraud, in particular by wealthy individuals and business enterprises operating or domiciled in the State party's jurisdiction, including through the adoption and enforcement of mandatory due diligence mechanisms, in order to contribute to international efforts to that effect and to enable other countries to secure the resources necessary for the realization of economic, social and cultural rights; (d) Take all measures necessary to avoid a situation that allows for shell companies to be used for profit-shifting, tax evasion and fraud by, inter alia, strengthening its legal framework and measures for the protection of whistle-blowers; (e) Conduct an independent and comprehensive assessment of the impacts of its national and international tax policy on the economies of developing countries and report on the findings in its next periodic report" (para 15c,d,e). |

*(Continued)*

**Table 2.** (Continued)

| Year | Country, where relevant | UN body or expert | Quotation or statement |
|---|---|---|---|
| 2024 | | The Independent Expert on the effects of foreign debt and other related international financial obligations of States on the full enjoyment of all human rights, particularly economic, social and cultural rights, Attiya Waris (A/HRC/55/54) [57] | The Independent Expert, in her annual report titled "Fiscal legitimacy through human rights: a principled approach to financial resource collection and allocation for the realization of human rights", "focuses on fiscal legitimacy and the import and requirements of a principle-based approach to the use of financial resources in the realization of human rights" (p 1). In particular, "International, binational or regional regulation is crucial for efficiency in combating tax evasion, avoidance or fraud and illicit financial flows. States are responsible for carefully examining different policy options at any and all times and for determining the most appropriate measures, not only in the light of their circumstances, but also considering their international and domestic human rights obligation" (para 36). |
| 2024 | Liechtenstein | The Independent Expert on the effects of foreign debt and other related international financial obligations of States on the full enjoyment of all human rights, particularly economic, social and cultural rights, Attiya Waris (A/HRC/55/54/Add.1) [58] | In the Independent Expert's report published in 2024 on her visit to Liechtenstein in 2023: "In the context of the financial and banking sector, the Independent Expert wishes to emphasize the need to assess the impact of investments, banking and financial activities outside Liechtenstein and to consider the extraterritorial human rights obligations of the State and businesses. She invites the State and businesses to apply a human rights lens to their activities and their international investments so as to avoid violations and abuses of human rights. Therefore, codes of conduct, sustainability strategies and other reference documents used by private actors, banks and investors should be fully in line with human rights and the Guiding Principles on Business and Human Rights: Implementing the United Nations "Protect, Respect and Remedy" Framework. In that context, they may seek advice from the Working Group on the issue of human rights and transnational corporations and other business enterprises" (para 91). |
| 2024 | | UN Committee on Economic, Social and Cultural Rights (Draft prepared by the Committee, open for public comment at the time of journal publication) [59] | In its draft General Comment on Economic, Social and Cultural Rights (ESCRs) and the Environmental Dimension of Sustainable Development, the Committee expounds on "Extraterritorial obligations and business entities" (para 33-39). This includes, "The Committee has previously affirmed that States have an obligation to respect, protect, and fulfil human rights for all, including when national policies affect those outside their territories. States may be in breach of these obligations if they fail to: prevent foreseeable harm to ESCRs resulting from climate change, environmental degradation, or unsustainable development; to regulate activities of private and public actors contributing to such harm; and to mobilize the maximum available resources in an effort to do so" (para 33). "States must create an international enabling environment conducive to the universal fulfilment of ESCRs, including in matters relating to bilateral and multilateral trade, investment, taxation, finance, environmental protection, climate change, and development cooperation. States must assess the risks and potential extraterritorial impacts of their laws, policies, and practices on the enjoyment of ESCRs, including as a result of environmental harm. Where a State's activities cause harm to the global commons that in turn threatens or harms Covenant rights, including through activities that cause climate change, the State has an extraterritorial obligation to respect and protect ESCRs through preventing such harm and providing appropriate compensation for loss and damage" (para 34). On cross-border tax, the Committee emphasises "The sustainable enjoyment of ESCRs is threatened by tax evasion, illicit financial flows, environmental crimes, and global corruption. States therefore have an obligation to protect Covenant rights through measures addressing these challenges, including international cooperation and appropriate regulation and legislation to prevent tax abuse, financial secrecy, and permissive corporate taxation inconsistent with the obligation to mobilise the maximum of available resources for the fulfilment of ESCRs" (para 35). |

tax avoidance strategies in the countries concerned" [44]. Other state practices, including lowering tax rates to encourage investment inducing a race to the bottom in tax rates, banking secrecy, were described as inconsistent with the duties of states under the covenant as they undermine the rights of all states to mobilise maximum resources for development. In January 2025, the Committee closed its a public call on a draft General Comment on Economic, Social and Cultural Rights and the Environmental Dimension of Sustainable Development, reiterating extraterritorial obligations of states "to prevent tax abuse, financial secrecy, and permissive corporate taxation inconsistent with the obligation to mobilise the maximum of available resources for the fulfilment of ESCRs [economic, social and cultural rights]" [59].

In 2014, the UN Special Rapporteur on extreme poverty and human rights dedicated their annual thematic report to the role of taxation [42]. This was a critical juncture in highlighting the crucial connection between extraterritorial obligations on taxation and financial transparency, and the enjoyment of human rights, drawing on international human rights law: "With regard to international cooperation and extraterritorial impact, each State should refrain from any conduct that impairs the ability of another State to raise revenue as required by their human rights commitments, and cooperate in creating an international environment that enables all States to fulfil their human rights obligations" (para 80). Since then, former and current mandate holders of the office of the Independent Expert on the effects of foreign debt and other related international financial obligations on the full realisation of human rights particularly economic, social, and cultural rights have emphasised the importance of the state's extraterritorial human obligations to address tax abuse and illicit financial flows. Consequently, as Table 2 shows, three UN committees have underscored the negative impact of cross-border tax abuse on the realisation of human rights and the extraterritorial obligations of the state. They have consistently called upon European tax havens to reform domestic law to ensure the rights of citizens' elsewhere may be realised and are not undermined by tax abuse [40]. For example, the UN Committee on the Elimination of Discrimination against Women has requested Cyprus in 2018 [45], Luxembourg in 2018 [47], New Zealand in 2018 [63], Panama in 2022 [46], Switzerland in 2016 and 2022 [64,43], and the UK and Northern Ireland in 2019 [49] to undertake impact assessments of their financial secrecy and corporate tax policies on women's rights and equality. Similarly, the UN Committee on the Rights of the Child, in its review of Ireland in 2020 and 2023 [50,55] and the Netherlands in 2022 [53], recommended that the countries ensure that their tax policies "do not contribute to tax abuse by national companies operating outside the State party that lead to a negative impact on the availability of resources for the realisation of children's rights in the countries in which they are operating" [53].

Most recently, in 2024, the UN Committee on Economic, Cultural and Social Rights made the most extensive recommendations in its concluding observations to Ireland on its fourth periodic review, where it noted its concerns "about reports that financial secrecy legislation and permissive corporate tax rules continue to hinder the ability of the State party, as well as other States, to meet their obligation to mobilize the maximum available resources for the implementation of the rights enshrined in the Covenant" [56]. The Irish Human Rights and Equality Commission also echoed the Committee's call for Ireland to assess the cross-border impact of domestic policy and revise it according to findings. This is significant as it is the first time a national human rights institution in a European tax haven has made recommendations to its government to identify "tax structures which are impacting rights protections in other territories", and undertake "ongoing independent and regular assessments of the impact of its policies on cross-border tax abuse" [65]. These and other international and regional human rights bodies justify extraterritorial human rights obligations of states at the doctrinal level both on legal and moral grounds. This is not yet accepted as an international human rights norm [66–68], and as such states may justify non-compliance, including with recommendations made by international human rights bodies [62]. Indeed, "the idea of invoking socioeconomic human rights obligations against states other than the territorial states […] isn't] universally accepted within the academy […] Moreover, governments in most advanced economies have been unwilling to recognise legally binding socio economic ETOs [extraterritorial obligations], given their potential to require the redistribution of states' resources" [69]. Nevertheless, as the evolution in Table 2 reflects, there is growing recognition that when a country acts through law, policy or practice in ways that foreseeably affect human rights in other jurisdictions, it must respect and protect those rights.

## The international economic environment

States are generally seen as responsible for upholding and protecting rights. According to the UN General Assembly, each country has primary responsibility for its own economic and social development [21]; thus, governments are obliged to respect, protect, and fulfil human rights at all times in their fiscal policies. Economic reforms that lead to retrogression should be avoided, even in extreme economic conditions [70]. While states are the duty bearers for their citizens' rights, a state's policy decisions, revenue, governance, and capacity to provide public services can be influenced by the actions and policies of other countries, global institutions and multinational companies. Consequently, the UN General Assembly in adopting the Addis Ababa Agenda for Action emphasised that national development efforts must operate within an enabling international economic environment, which includes supportive global trade, monetary and financial systems, as well as improved global economic governance [71].

There are multiple shapers of the international economic environment. The policies and positioning of high-income countries and the Bretton Woods Institutions are particularly influential. At the same time, multinational companies can influence global rules by supporting policies such as trade liberalisation, impeding regulations like those concerning the environment and labour, and contributing to regulations like through standards in global supply chains [72]. This influence is evident in the strong rules and regulations surrounding trade and intellectual property and in the less robust rules around tax collection, workers' rights, and environmental protection [73]. These influences are compounded by shortcomings in cooperation at the international level in taxation—the focus of this paper—and in mechanisms for resolving sovereign debt crises efficiently and fairly [74].

The main challenges stemming from gaps in global governance on tax cooperation include [74]: 1) taxing companies and individuals operating and with assets in multiple tax jurisdictions, 2) the race to the bottom in income tax rates associated with the mobility of capital, 3) profit shifting by multinational companies which undermines tax collection, and 4) the existence of tax havens, which facilitate illicit financial flows.

The most recent estimates suggest multinational companies shift US$1.42 trillion in profits into tax havens each year, which causes governments around the world to lose US$348 billion a year in direct tax revenue [75]. This global aggressive corporate tax avoidance is facilitated by the vulnerabilities states create through domestic tax policy and their influence on international tax rules. Large-scale cross-border profit shifting by multinational companies enabled by OECD members, their dependencies, and other tax havens severely undermines the ability of other governments to raise revenue for providing quality services to fulfil core social and economic rights for its people, as is the case in Nigeria. Over two-thirds of all global tax losses can be attributed to OECD members and their dependencies [75]. As the results show, profit shifting weakens governance and causes significant deprivations in human rights and the loss of life.

To address these challenges in the context of international tax cooperation, international tax rules have been set by the OECD and its 38 members, the world's highest-income countries. While some non-OECD members have entered discussions under the OECD/G20 Inclusive Framework on Base Erosion and Profit Shifting initiative—particularly on how taxing the digital economy and combating tax avoidance by multinational companies—the OECD's mandate is to protect the interests of its member states. Among the 38 OECD member countries, 27 are European nations, and of these, 22 are members of the EU.

The most recent tax reforms championed by the OECD, known as the "two-pillar solution", have not been accompanied by the OECD publishing detailed country-level revenue estimates or human rights impact assessments. Independent assessments indicate that revenue gains in the Global South may be significantly lower than in the Global North; some countries might

even lose revenue [76,77]. For example, an assessment by the EU Tax Observatory on the revenue implications of Pillar One Amount A of the OECD's reforms, which seeks to reallocate the profits of large multinationals to market countries, finds that "developed countries would collect more than 77% of the net revenues […] [and] in absolute terms, the least developed countries do not benefit much from Amount A, as their net revenues are almost null" [78]. The UN Secretary-General has described these rules as having "limited effectiveness" [79].

Due to these concerns, as part of Special Procedures of the UN Human Rights Council, a group of UN Independent Experts with human rights mandates has called on the OECD to assess the human rights impacts of the two-pillar solution, including gender and racial implications. They have raised concerns about the lack of an "effective voice" and voting rights for non-OECD/G20 countries in designing these solutions, as well as their limited scope, which only covers a portion of the global profits of around 100 of the largest multinational companies under Pillar One [80]. The experts highlighted potential harmful effects on the taxing rights of countries in the Global South, such as restrictions on unilateral taxes on digital services, and reduced revenue collection "and consequently on the availability of resources for the progressive realisation of economic, social, and cultural rights", which could have gender, ethnic, and racial discriminatory impacts [81].

No African countries are members of the OECD, which is hosted in Paris, and fewer than half have chosen or been able to join the Inclusive Framework. For those that do participate, the UN Secretary-General reports significant barriers that prevent "their meaningful engagement in agenda-setting and decision-making" [79]. As a result, "it often happens that the substantive rules developed through these OECD initiatives do not adequately address the needs and priorities of developing countries and/or are beyond their capacities to implement" [79].

This concern has been a primary driver for all African nations, as part of the Africa Group at the UN, to propose a resolution in late 2023 aimed at promoting inclusive and effective international tax cooperation at the UN, which has the legitimacy to be a truly inclusive and effective forum [79]. The ambitions of the resolution include establishing a UN framework convention on tax, recognising taxation's crucial role in closing the financing gap for sustainable development. The resolution states, "so that Governments may better cooperate in generating financing for development, [...] Noting the corrosive effect that aggressive tax avoidance and tax evasion have on trust, the social compact, financial integrity, the rule of law and sustainable development, affecting the poorest and most vulnerable" [82]. This is critically important, especially for countries that urgently need revenue for development and to adapt to climate emergencies.

## International tax reforms

On 22 November 2023, UN member countries voted on the resolution (A/C.2/78/L.18/Rev.1). A total of 125 countries voted in favour of inclusive tax cooperation at the UN, 48 countries voted against it, and 9 countries abstained, making a total of 182 votes [83,84]. Notably, all EU member states voted against the resolution, Norway abstained, while Belarus and Russia voted in favour. In the aforementioned letter to the OECD, the UN Independent Experts stated, "The negotiations related to the newly adopted resolution on the UN Convention on International Tax Cooperation, and the positions that some OECD member states under the aegis of the UN could undermine the development of more effective multilateral standards for strengthening international tax cooperation" [80]. In contrast, all the African countries that voted supported the resolution.

Most European countries support the OECD two-pillar solution and hinder international tax reform efforts at the UN. In August 2024, during the vote on the terms of reference for the UN Framework Convention on International Tax Cooperation, all EU member states

abstained, and eight OECD nations voted against the terms of reference, including the USA and UK [85]. Yet the EU—of which most European nations are members—is founded on the principles of human rights within and outside its borders. The treaty that established the EU alongside its Charter of Fundamental Rights also guides its external action:

> In its relations with the wider world, the Union shall uphold and promote its values and interests and contribute to the protection of its citizens. It shall contribute to peace, security, the sustainable development of the Earth, solidarity and mutual respect among peoples, free and fair trade, eradication of poverty and the protection of human rights, particularly the rights of the child, as well as to the strict observance and the development of international law, including respect for the principles of the United Nations Charter. [86]

European citizens support their governments in ensuring that they do not undermine the fulfilment of rights globally and that European companies operating abroad do not harm human rights. A 2021 YouGov poll showed over 80% support among nine EU member states for the EU to introduce laws to hold companies operating outside the EU accountable for any human rights violations and environmental harms [87].

## Nigeria

Nigeria is one of the African countries with the highest illicit financial flows, according to a landmark 2015 report written by the High Level Panel on Illicit Financial Flows from Africa, which was a panel tasked by the African Union Commission and the UN Economic Commission for Africa to address illicit financial flows and make recommendations on how to curb them [88]. The Nigerian government, as part of its Integrated National Financing Framework, which is the national level sustainable financing plan to achieve the SDGs, identifies the primary contributors to illicit financial flows as "aggressive corporate tax avoidance, abusive transfer pricing, and trade mis-invoicing" [89]. Although Nigeria has a low tax-to-GDP ratio (6.7% compared to the African average of 15.6% in 2021), corporate income tax accounts for the largest share of tax revenues (35%) [90].

Unlike European countries, Nigeria has been a resolution sponsor and consistently voted in favour of efforts for international tax cooperation at the UN. While all EU member states supported the OECD's two-pillar solutions, as a member of the OECD/G20 Inclusive Framework, Nigeria did not endorse the two-pillar solution, citing concerns over reduced revenue given that most multinational companies operating in Nigeria do not meet the group-level turnover threshold, as well as flagging the risk of agreeing to international arbitration which may yield limited returns but come with high costs [91]. The country's Integrated National Financing Framework stated, "The proposals and frameworks designed by the OECD and the G7 to address issues related to IFFs [illicit financial flows], base erosion and profit shifting are complex and, in some cases, unclear if they are beneficial to developing countries like Nigeria. This is one of the reasons why Nigeria recently rejected the G20 and G7 proposal on a global minimum tax rate even though 130 countries agreed to it". Nigeria is now considering the implementation of pillar two on a global minimum tax of 15% or introducing a Qualified Domestic Minimum Top-up Tax while reviewing tax incentives to ensure Nigeria does not cede its tax base to other countries and to protect national interests, according to the Federal Inland Revenue Service [92].

Given the importance of corporate tax revenue and the scale of profit shifting (26% of corporate tax revenue with almost one-quarter to European tax havens [93]), Nigeria makes an interesting case study to examine the impacts of profit shifting on rights.

## Aims

This study explores the impact of cross-border aggressive corporate avoidance—or tax abuse—on human rights. It focuses on three main areas:

1) The impact of profit shifting from Nigeria to tax havens, including European tax havens, on rights and governance in Nigeria.

2) The impact of increased revenue from profits shifted out of Nigeria on rights in European tax havens.

3) The impact of the total gains from all shifted profits on rights in European tax havens.

This research is novel because, while previous studies have analysed the effect of profit shifting on rights in those jurisdictions which lose revenue [28,37,94], this study also analyses the gains in terms of rights in tax havens.

## Method

### Step 1. Tax revenue lost and gained from profit shifting

In the first step, we present estimates of the revenue lost from Nigeria and the revenue gained by European tax havens, generated from taxes on profits shifted.

Two widely accepted methods exist for quantifying the scale of profits shifted to tax havens: the approaches by Garcia-Bernardo and Janský [95] and by Tørsløv, Wier and Zucman [96].

The Garcia-Bernardo and Janský approach, used in the annual *State of Tax Justice* report published by the Tax Justice Network, is termed the "misalignment" approach [95,97]. This method analyses aggregate anonymised country-by-country reporting data published by the OECD and the US. It compares differences between the reported profits of multinational companies and theoretical profits, attributing differences to profit misalignment. Theoretical profits are those expected based on economic activity, which can be measured using proxies, such as a combination of tangible assets, number of employees, and/or wages.

In contrast, Tørsløv, Wier and Zucman's method, used in the EU Tax Observatory's *Global Tax Evasion Report 2024* [19,96], estimates profit shifting and then allocates these shifted profits back to source countries. In their latest estimates published by Wier and Zucman in 2022, they define profit shifting "as a tax-motivated and artificial transfer of paper profits within a multinational firm from high-tax countries to low-tax locales" [98]. They measure "profit shifting to tax havens as the amount of multinational profits booked by companies in these havens above and beyond what can be explained by real economic activity", which includes capital, labour, research and development spending [98]. Tørsløv, Wier and Zucman assess the profits-to-wage ratio of foreign and local firms in tax havens, non-tax haven OECD countries, and large developing countries. Generally, they find that foreign firms in tax havens have much higher profits-to-wage ratio than local firms, while in non-tax havens, foreign firms tend to be slightly less profitable than local firms. The authors state that the excess profitability of foreign subsidiaries in tax havens is explained by above-normal intra-group transfers from high- to low-tax jurisdictions. To estimate the profits shifted to each tax haven, they assume that, without profit shifting, the profits-to-wage ratio of foreign firms would be the same as for local firms. Second, they use bilateral balance of payments data to trace the origins of these payments and allocate the excess haven profits to the countries from which the profits were shifted, thereby estimating the resulting tax losses of the source countries.

A key distinction between the two methodologies relevant for this paper is that Tørsløv et al. present estimates for the profits shifted from source countries to destination tax havens, including the resultant losses and gains (e.g. Nigeria's losses due to profits shifted to the

Netherlands and the Netherlands' gains from profits shifted out of Nigeria). Conversely, Garcia-Bernardo and Janský's estimates quantify the scale of profits shifted inward and outward but do not present data on the bilateral relationships between jurisdictions. For our study, which aims to understand the cross-border impacts and responsibilities for profit shifting and revenue losses, we use Tørsløv et al.'s data, as it includes bilateral information.

Wier and Zucman's most recent estimates, based on their methodology developed with Tørsløv, detail the corporate tax revenue that jurisdictions gained and lost due to profit shifting. This dataset is available on the Missing Profits website and for download [93,98,99]. The data allows observation of one country's gains from the tax on profits shifted from another country, and the losses sustained from foregone tax revenue as a result of profits being shifted out. Their estimates (as seen in Table C4 of the downloadable dataset "1975-2019 updated estimates: Tables" [93]) list profits shifted from origin countries to tax havens in 2019.

Nigeria is one of the two African countries—South Africa being the other—for which detailed data is available in the dataset, and we chose Nigeria as an interesting case study as explained above. According to Wier and Zucman, in 2019, Nigeria lost 26% (South Africa loses 13%) of its corporate tax revenue to tax havens. Data is available concerning losses attributed to EU havens (specifically, Belgium, Cyprus, Ireland, Luxembourg, Malta, and the Netherlands) and non-EU tax havens (Switzerland and others, which are aggregated) [93]. We analyse the potential of additional government revenue for Nigeria equivalent to the revenue lost to all tax havens and to European tax havens, if profits were taxed at Nigeria's statutory corporate income tax rate of 30%. We rounded this figure to the nearest million. Profits shifted to two jurisdictions—Cyprus and Malta—resulted in less than US$0.5 million in corporate income tax revenue loss in Nigeria and have been excluded.

The authors categorise countries as tax havens if foreign firms have excessive profitability (in comparison to local firms) and the effective corporate tax rate is below 15%. The limitations of this approach to categorising tax havens is discussed below. Additionally, the dataset includes estimates of the tax rate on shifted profits into European tax havens in 2018; we use the rates provided on the Missing Profits website [99], which are slightly lower for Ireland and Luxembourg than those available for download [93]. We use this information to calculate the tax revenues gained by European tax havens from profits shifted inward from all jurisdictions, and specifically from Nigeria.

## Step 2. Human rights and governance impacts of profit shifting

In the second step, we use the Government Revenue and Development Estimations (GRADE) model (version V3.12.2: 2024/10/17) to translate revenue gains and losses into indicators of access to several rights, including impacts on governance. The model employs four decades of data to analyse the effects of changes in government revenue in any given country. Since an increase in revenue takes time to show impact, GRADE assumes that it takes five years for revenue increases to affect rights coverage, which then gradually plateaus over the longer term.

The research underpinning GRADE employed unbalanced panel data modelling for 191 countries from 1980 to 2022, expressing access to rights as percentages ranging from 0 to 100. The data is considered unbalanced because many countries lack complete data across the four decades. GRADE uses the 2023 Government Revenue Database [100]. It also uses the Worldwide Governance Indicators, which are six quality of governance indicators: Control of Corruption, Government Effectiveness, Voice and Accountability, Political Stability, Rule of Law, and Regulatory Quality, ranging from -2.5 to +2.5 [101]. For the rights variables, the World Bank World Development Indicators database is used to model government revenue and the coverage of water and sanitation (the percentage of the population with access to

basic and safely managed water and sanitation) [102]. It also models government revenue and school attendance using out-of-school data from the UNESCO VIEW dataset [103], and school-age population statistics from the UNESCO Institute for Statistics (UIS) [104]. All databases are updated annually except the World Development Indicators, which are updated twice each year. The GRADE dataset is updated annually, in October, using the most recent data available.

As explained in previous work [17], a similar non-linear structure of the following form is used for the basic model of each sector.

$$(Y_i) = 1 / (1 + e^{-((\alpha + \chi w)(\log(GR) - (\beta + \delta w))})$$

Here, $Y_i$ is the specific rights variable being modelled, GR is government revenue, $\omega$ is a vector of the quality of governance indicators for each country over time, and $\alpha \chi \beta$ and $\delta$ are the parameters to be estimated. This non-linear logistic function was chosen to mimic the real-world development of many of these rights indicators. A linear relationship between revenue and these variables would be inappropriate, as a standard panel logistic function would impose the same 'S' shaped curve on all countries (see Fig 1). The logistic function parameters have been augmented with the Worldwide Governance Indicator dataset, which allows each country to have a different 'S' shape as its governance quality varies [105].

This S-shaped curve (see Fig 1) illustrates the typical relationship between government revenue per capita and rights coverage. At the lower end of the curve, when government revenue per capita is minimal, increases in revenue will have little impact on the coverage of rights (the dependent variable). This is followed by a rapid increase in coverage of rights as revenue

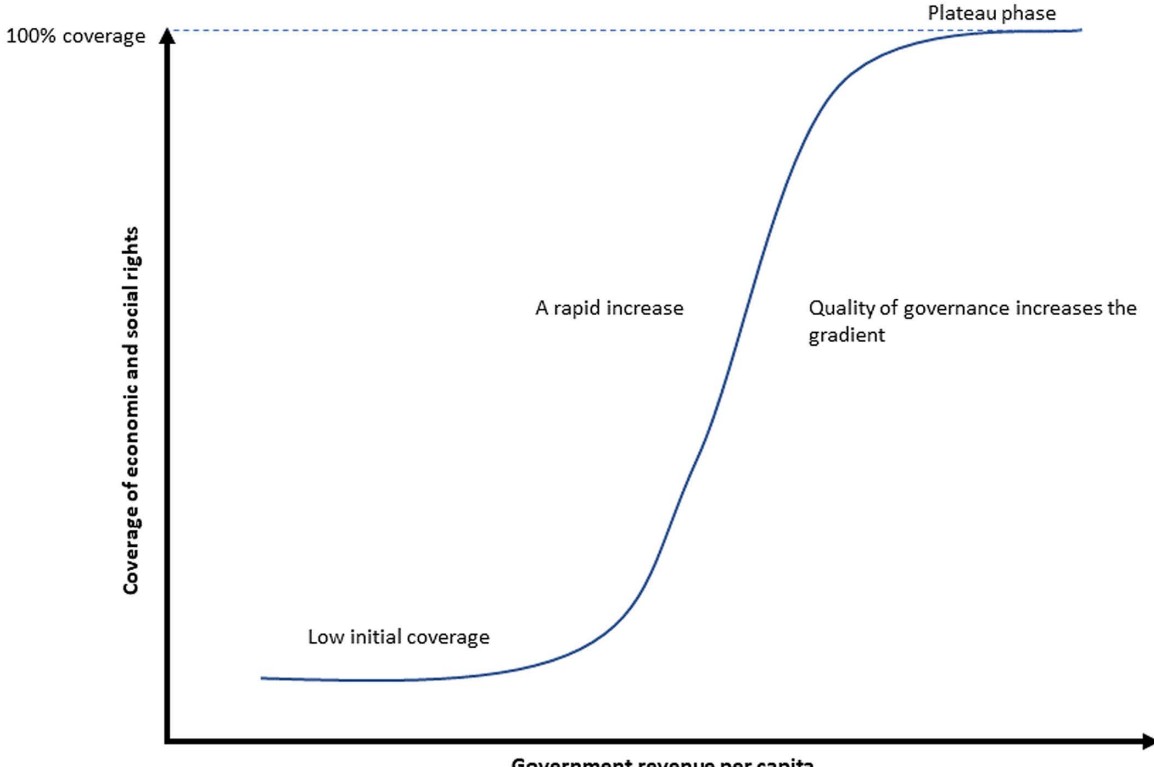

**Fig 1. Relationship between government revenue per capita and coverage of rights.**

per capita increases. Finally, the S-shaped curve tapers off or plateaus when almost 100% of the population has access to these rights in a country, meaning that additional revenue yields minimal improvements. Governance indicators also respond to increased government revenue. The GRADE model uses two contrasting econometric methodologies to quantify the effects of increasing government revenue per capita on indicators of governance quality, showing that increased government revenue significantly influences governance indicators. There is an important feedback loop between government revenue and governance; over time, as governance improves, government revenue increases further, which further improves governance, creating a virtuous cycle [28]. Thus, using a similar set of equations for the quality of governance indicators, the model can realistically estimate the impact of a change in government revenue on the six Worldwide Governance Indicators and rights.

The quality of governance is critical and changes the shape of the S-shaped curve, making it unique to each country (see Fig 1). Therefore, additional revenue and small improvements in governance can greatly impact lower-income countries. In contrast, the coverage of rights approaches 100% in European countries, so additional revenue has little effect.

Importantly, we do not assume that a government will allocate new revenue differently than it has in the past. Many prior studies on government expenditure unrealistically assume that governments will spend all additional revenue on a specific activity, such as a school feeding programme or nurses' salaries. In reality, governments have competing priorities and typically do not dedicate additional revenue to a single activity or budget line. Therefore, the GRADE model operates under the assumption that any increase in government revenue will be allocated in a manner consistent with past practices. This means that additional revenue and improved governance impact all social and economic outcomes, not just those highlighted in this study.

Using the GRADE model, we expressed revenue losses and gains from step one as percentages of total government revenue. We projected the impact of these on rights over the years 2002 to 2022, acknowledging the time it takes for the impact on rights to materialise. The results are presented as the absolute number of additional people who access their rights in one year, typically the final year of 2022, depending on data availability. We analysed the six Worldwide Governance Indicators and present here the improvement in two of these indicators: Control of Corruption and Government Effectiveness. These indicators do the most work in the models and are most greatly affected by increases in government revenue.

## Results and limitations

### Results

In 2019, Nigeria lost US$1,291 million in tax revenue (equivalent to US$1,203 million in 2015 USD), which accounts for 3.02% of government's total revenue, due to all tax havens included in Wier and Zucman's dataset [93]. Nearly one-quarter of these lost revenues went to EU tax havens, with the largest tax loss to the Netherlands at US$207 million. The remaining US$970 million went to non-EU tax havens including Switzerland, Bermuda, the Caribbean, Puerto Rico, Hong Kong, Singapore, and others.

Our modelling indicates that if the Nigerian government had additional revenue equivalent to these losses, an additional 500,000 Nigerians would have their right to drink clean water and nearly 800,000 their right to use basic sanitation each day, 150,000 Nigerian children could exercise their right to education and 11 children could have their right to survive each day (totalling 4,063 children each year).

The rights deprivations attributed to the tax havens' policies in proportion to their contributions to the losses from Nigeria are detailed in Table 3. Additionally, our modelling

**Table 3. Human rights deprivations in Nigeria and the role of tax havens.**

| | Tax revenue losses/gains ($, millions) 2019 | Total losses (%) | Basic water (SDG 6) | Safe water (SDG 6) | Basic sanitation (SDG 6) | U-5 survival (SDG 3) | Maternal survival (SDG 3) | Primary school attendance | Lower secondary school attendance | Upper secondary school attendance | Total school attendance (SDG 4) |
|---|---|---|---|---|---|---|---|---|---|---|---|
| From Nigeria to all havens | 1291 | 100.00 | 517,759 | 116,657 | 796,568 | 4,063 | 263 | 72,618 | 38,523 | 39,139 | 150,279 |
| | | | **Numbers deprived of their rights in Nigeria attributable to tax havens** | | | | | | | | |
| EU tax havens | **321*** | **24.86** | 128,738 | 29,006 | 198,062 | 1,010 | 66 | 18,056 | 9,578 | 9,732 | 37,366 |
| Belgium | 33 | 2.56 | 13,235 | 2,982 | 20,362 | 104 | 7 | 1,856 | 985 | 1,000 | 3,841 |
| Ireland | 51 | 3.95 | 20,454 | 4,608 | 31,468 | 160 | 10 | 2,869 | 1,522 | 1,546 | 5,937 |
| Luxembourg | 29 | 2.25 | 11,631 | 2,621 | 17,893 | 91 | 6 | 1,631 | 865 | 879 | 3,376 |
| Netherlands | 207 | 16.03 | 83,018 | 18,705 | 127,722 | 651 | 42 | 11,644 | 6,177 | 6,275 | 24,096 |
| Non-EU tax havens | **970** | **75.14** | 389,021 | 87,651 | 598,506 | 3,052 | 198 | 54,562 | 28,944 | 29,407 | 112,913 |
| Switzerland | 10 | 0.77 | 4,011 | 904 | 6,170 | 31 | 2 | 562 | 298 | 303 | 1,164 |
| Bermuda, Caribbean, Puerto Rico, Hong Kong, Singapore and others | 960 | 74.36 | 385,011 | 86,748 | 592,336 | 3,021 | 196 | 53,999 | 28,646 | 29,104 | 111,749 |

* Due to rounding, the figures not emboldened may not always add up to emboldened totals from 'EU tax havens' and 'Non-EU tax havens'.

**Table 4. Changes in governance indicators in Nigeria with additional revenue equivalent to that lost to tax havens.**

| Year | Government effectiveness without additional revenue | Government effectiveness with additional revenue | Control of corruption without additional revenue | Control of corruption with additional revenue |
|---|---|---|---|---|
| 2002 | −1.0197 | −1.0197 | −1.5021 | −1.5021 |
| 2003 | −0.9161 | −0.9148 | −1.4177 | −1.4165 |
| 2004 | −0.9356 | −0.9334 | −1.3834 | −1.3814 |
| 2005 | −0.8972 | −0.8943 | −1.1808 | −1.1782 |
| 2006 | −0.9675 | −0.9641 | −1.1264 | −1.1233 |
| 2007 | −1.0321 | −1.0283 | −1.0669 | −1.0635 |
| 2008 | −0.9875 | −0.9835 | −0.9009 | −0.8973 |
| 2009 | −1.2005 | −1.1963 | −1.0419 | −1.0381 |
| 2010 | −1.1652 | −1.1609 | −1.0515 | −1.0476 |
| 2011 | −1.1022 | −1.0979 | −1.1894 | −1.1853 |
| 2012 | −1.0010 | −0.9965 | −1.1759 | −1.1718 |
| 2013 | −0.9977 | −0.9932 | −1.2269 | −1.2227 |
| 2014 | −1.1891 | −1.1846 | −1.2835 | −1.2793 |
| 2015 | −0.9962 | −0.9917 | −1.0957 | −1.0915 |
| 2016 | −1.1199 | −1.1154 | −1.0451 | −1.0409 |
| 2017 | −1.0414 | −1.0369 | −1.0998 | −1.0955 |
| 2018 | −1.1182 | −1.1137 | −1.0836 | −1.0793 |
| 2019 | −1.2133 | −1.2088 | −1.1151 | −1.1109 |
| 2020 | −1.1442 | −1.1397 | −1.1209 | −1.1166 |
| 2021 | −1.0278 | −1.0232 | −1.1033 | −1.0990 |
| 2022 | −1.0351 | −1.0305 | −1.1010 | −1.0967 |
| Improvement in governance indicators if the government had additional revenue equivalent to that lost to tax havens | | 0.004577 | | 0.004293 |

**Table 5. Gains in human rights in European tax havens from shifted profits from Nigeria.**

| | Nigerian profits lost ($, millions) 2019 | Tax rate on shifted profits (%) (Missing Profits) | Tax revenue gained from profits shifted from Nigeria ($, millions) 2019 | Tax revenue gained ($, millions in 2015) | % of government revenue in 2019 | Basic water (SDG 6) | Safe water (SDG 6) | Basic sanitation (SDG 6) | Safe sanitation (SDG 6) | U-5 survival (SDG 3) | Maternal survival (SDG 3) | Primary school attendance | Lower secondary school attendance | Upper secondary school attendance | Total School Attendance (SDG 4) |
|---|---|---|---|---|---|---|---|---|---|---|---|---|---|---|---|
| All havens | **4303** | | | | | | | | | | | | | | |
| EU tax havens | **1069** | | | | | | | | | | | | | | |
| | | | | | | Human rights gains in European tax havens attributable to profits shifted inward from Nigeria | | | | | | | | | |
| Belgium | 112 | 20 | 22.40 | 20.88 | 0.01 | 0 | 0 | 0 | 173 | 0 | 0 | 0 | 0 | 2 | 2 |
| Cyprus | 1 | 7 | 0.07 | 0.07 | 0.00 | 0 | 0 | 0 | 0 | 0 | 0 | 0 | 0 | 0 | 0 |
| Ireland | 170 | 6 | 10.20 | 9.51 | 0.01 | 0 | 0 | 0 | 48 | 0 | 0 | 0 | 0 | 0 | 0 |
| Luxembourg | 97 | 4 | 3.88 | 3.62 | 0.01 | 0 | 0 | 0 | 0 | 0 | 0 | 0 | 0 | 0 | 0 |
| Netherlands | 689 | 6 | 41.34 | 38.53 | 0.01 | 0 | 0 | 0 | 173 | 0 | 0 | 0 | 0 | 2 | 2 |
| Non-EU tax havens | **3233** | | | | | | | | | | | | | | |
| Switzerland | 33 | 8 | 2.64 | 2.46 | 0.00 | 0 | 0 | 0 | 0 | 0 | 0 | 0 | 0 | 0 | 0 |
| Total | | | | | | **0** | **0** | **0** | **394** | **0** | **0** | **0** | **0** | **4** | **4** |

**Table 6. Gains in human rights in European tax havens from *all* shifted profits.**

| | Tax revenue gained by tax havens from all profits shifted inward ($, billions) 2019 | Tax revenue gained by tax havens from all profits shifted inward ($, billions) 2015 | % of government revenue in 2019 in USD 2015 | Basic water (SDG 6) | Safe water (SDG 6) | Basic sanitation (SDG 6) | Safe sanitation (SDG 6) | U-5 survival (SDG 3) | Maternal survival (SDG 3) | Primary school attendance | Lower secondary school attendance | Upper secondary school attendance | Total school attendance (SDG 4) |
|---|---|---|---|---|---|---|---|---|---|---|---|---|---|
| EU tax havens | | | | | | | | | | | | | |
| | | | | Human rights gains in European tax havens attributable to profits shifted inward from all countries | | | | | | | | | |
| Belgium | 7.60 | 7.08 | 2.87 | 0 | 0 | 0 | 48,186 | 0 | 0 | 200 | 63 | 453 | 716 |
| Cyprus | 0.30 | 0.28 | 2.89 | 0 | 0 | 0 | 6,357 | 1 | 0 | 0 | 13 | 41 | 54 |
| Ireland | 7.20 | 6.71 | 7.3 | 0 | 0 | 0 | 32,373 | 1 | 0 | 0 | 0 | 0 | 0 |
| Luxembourg | 2.40 | 2.24 | 7.41 | 0 | 0 | 0 | 217 | 0 | 0 | 10 | 5 | 30 | 45 |
| Netherlands | 6.50 | 6.06 | 1.64 | 0 | | 0 | 27,898 | 0 | 0 | 0 | 0 | 267 | 267 |
| Non-EU tax havens | | | | | | | | | | | | | |
| Switzerland | 8.90 | 8.30 | 3.21 | 0 | 0 | 0 | 11,231 | 0 | 0 | 0 | 48 | 228 | 276 |
| Total | | | | **0** | **6** | **0** | **126,262** | **2** | **0** | **210** | **129** | **1,019** | **1,358** |

suggests that if the government had this additional revenue equivalent to losses, there would be improvements in governance indicators. Table 4 presents two of these indicators: Control of Corruption and Government Effectiveness.

European countries tax shifted profits at relatively low rates; the Netherlands taxes shifted profits at 6%, Ireland at 6%, Luxembourg at 4%, and Cyprus at 7%, while Belgium has the highest rate of taxing shifted profits, set at 20% [99]. The tax revenue generated from profits shifted out of Nigeria generally accounts for just 0.01% of these European government's revenue.

Since the coverage of many rights approaches 100% in European countries, the effect of shifted profits from Nigeria is minimal (see Table 5). Attracting profits that should have been booked in Nigeria only enables around 400 people to access safe sanitation and guarantees 4 children the right to attend school in European tax havens.

When considering all inward-shifted profits, the financial impacts in these tax havens become more significant (see Table 6). For example, tax revenue from all shifted profits contributes 7.3% to government revenue in Ireland. However, since many rights are already widely covered in these countries, the additional revenue has little impact overall. Notably, though, our modelling indicates that it does enable 126,000 additional people to gain access to safe sanitation in Belgium, Cyprus, Ireland, Luxembourg, and the Netherlands.

## Limitations

There is no universally accepted definition of a tax haven, but generally, we broadly understand these as jurisdictions that allow non-resident multinational companies and individuals to pay lower tax where they are resident or operate [106,107]. The tax haven listed used by Wier and Zucman [98] is the same one used in their research with Tørsløv [96]. This list includes 41 territories and overseas territories, including five OECD countries: Belgium, Ireland, Luxembourg, Netherlands, and Switzerland. It is based on the tax haven list developed by Hines and Rice [108], with the addition of the Netherlands, Belgium, and Puerto Rico.

However, Wier and Zucman's dataset does not include disaggregated information for all of Nigeria's counterpart countries. It only includes disaggregated data for EU tax havens—namely Belgium, Cyprus, Ireland, Luxembourg, Malta, and the Netherlands—as well as Switzerland. Other non-EU tax havens included in the tax haven list have only aggregated data, which means a comprehensive analysis and discussion of all European jurisdictions and their dependencies, including the overseas territories and crown dependencies of the UK, such as Bermuda, the British Virgin Islands, and Jersey, is not possible. Additionally, four tax havens are ranked in the top 20 jurisdictions on the Corporate Tax Haven Index list [109]—an objective and verifiable tax haven list [106]—that are not included in the tax haven list Wier and Zucman use: the United Arab Emirates, the United Kingdom, France, and China. Furthermore, this study has focused on global tax abuse stemming from corporate profit shifting, while offshore wealth tax evasion could be included because it also undermines government revenue collection and the fulfilment of rights, facilitated by tax havens.

## Discussion

European tax havens significantly and negatively impact the Nigerian government's ability to raise resources to meet its human rights obligations and, importantly, citizens' right to good governance. In contrast, the gains that European tax havens make at Nigeria's expense are small compared to the overall government revenue, and the benefits to rights in Europe are very limited, given that coverage is already high. Europe is one of three major profit-shifting havens, alongside the Americas and Asia, with Ireland and the Netherlands being major destinations for profit shifting globally [19].

To illustrate this, if the Netherlands were to reform its domestic fiscal policy to halt profit shifting from Nigeria, nearly 130,000 additional Nigerians would gain access to basic sanitation (SDG 6) every day (Table 3). In contrast, this reform would not affect access to basic sanitation for any adults or children in the Netherlands (Table 5). Furthermore, reforming Dutch tax policies would lead to approximately 2 additional Nigerian under-five children surviving every day (651 each year), and just over 24,000 more Nigerian children attending school daily (Table 3). Although there would be a reduction in government revenue in the Netherlands due to these policy changes, it would not prevent any under-five deaths and would only impact the educational rights of two Dutch children (Table 5).

If the Netherlands implemented effective policies to stop inward profit shifting from all countries, it would sacrifice roughly 1.64% of its revenue (Table 6). This change would not impact the survival of any under-five children and would result in 267 children not having their right to education being met. Currently, the losses inflicted by the Netherlands severely undermine the maximum available resources in other countries, compromising the fulfilment of economic and social rights.

Profit shifting deepens inequalities between countries; while higher-income countries lose more in absolute numbers, lower-income countries suffer larger revenue losses as a proportion of their total revenue [95]. In these low-income countries, coverage of rights is much lower (Table 1). Profit shifting also deepens inequalities within countries. Less mobile taxpayers, including individual middle- and lower-income taxpayers and domestic businesses that do not utilise tax havens, generally pay more taxes relative to income. Government tax policies often focus on easier-to-administer taxes, which tend to be more retrogressive when applied to essential goods and services. The impact on lower-income households, who rely heavily on public services, is further exacerbated, as this study demonstrates, where reduced government revenue and weakened governance adversely impact public service provision essential for fulfilling rights.

## Intergovernmental cooperation and state duties

As explained in the background, European efforts as part of the OECD/G20 reforms on base erosion and profit shifting, which commenced in 2015, fall short (the second category in the categorisation of extraterritorial responsibilities described earlier [39]). Despite these reforms and reforms in the US, *The Global Tax Evasion Report 2024*, published by the EU Tax Observatory, states that "there was little discernible decline in profit shifting globally, or profit shifting by US multinationals relative to 2015", although profits shifted to tax havens match rather than outpace global foreign profits as was observed in the early 2010s. Even though the share of government revenue for European tax havens on all profits shifted inward ranges from 1.64% to 7.41% (Table 5), indicating that low tax rates can generate not insignificant tax revenues, the limited impact on rights of people living in these tax havens, but adverse effects on the international order to fulfil rights globally, necessitates much better tax rules, cooperation, and coordination.

European countries and the OECD are hindering efforts towards a UN Framework Convention on International Tax Cooperation, observed in efforts to weaken resolutions in 2022, 2023 and 2024 for negotiations on international tax cooperation and a framework convention, first proposed by Nigeria on behalf of the Africa Group at the UN. Absent a framework convention, unilateral state and EU-wide regulations are vital to creating an international order where rights can be achieved. These should be aligned with the 'ABCs' (automatic exchange of information, beneficial ownership transparency, and public country-by-country reporting) of tax transparency to aid governments in assessing and addressing tax abuse risks [110].

In line with the ABCs, Nigeria has introduced domestic regulations to implement the Common Reporting Standard of the Multilateral Competent Authority Agreement, the

OECD's limited standard for the automatic exchange of information between participating countries to share financial information to examine if individuals or corporations have earned income in other countries that has been hidden to underpay taxes; however, information asymmetries and rules may hinder Nigeria's access and use of relevant data. Beneficial ownership regulations for companies are in place, and despite loopholes, information is made public [111]. Nigeria has also signed the Multilateral Competent Authority Agreement to exchange country-by-country reports automatically and operationalised this domestically. Despite these efforts, as well as other regulations, initiatives, and new institutional arrangements, "IFFs [illicit financial flows] continue to thrive in the country" [89]. Recognising its own domestic governance challenges, the government also emphasises that existing efforts through the OECD, G7 and G20 to address base erosion and profit shifting are complex and may not necessarily benefit developing countries [89].

A UN Framework Convention on International Tax Cooperation needs to address the way multinational companies are taxed. At present, multinational companies are typically taxed using the arm's length principle, which artificially treats subsidiaries of multinational companies as separate entities for tax purposes. This facilitates extensive tax planning, so multinational companies may set up affiliates in tax havens to reduce taxes in countries with economic activity. In contrast, unitary tax through formulary apportionment considers the global profits of a multinational company since the company operates as a single unit in practice. Then for tax purposes, the global profits are apportioned between countries based on a combination of assets, employees, and/or sales [112,113]. This is where the multinational company has real business activities, as opposed, for example, to its subsidiary in a tax haven, set up with the primary purpose of shifting profits to jurisdictions with low or zero tax rates. Such international reforms would increase revenue towards the magnitude of estimated losses attributed to profit shifting; in Nigeria, the government's revenue would increase by 3.02%, and hundreds of thousands would access rights, as shown here.

In the short- and medium-term, to reduce profit shifting in the absence of a UN tax convention, the EU Tax Observatory suggests three critical areas to address global tax avoidance: 1) reopening negotiations on the global minimum corporate tax rate within the OECD's pillar 2 to remove carveouts and increase the rate from 15% to 25% to reduce tax competition and disincentivise profit shifting, 2) introduction of unilateral higher taxation of multinationals through a remedial tax based on the tax deficit of multinational companies where they pay less in other countries than the minimum set tax suggested as 25%, and 3) strengthening the application of anti-abuse rules to better prevent transactions that are intended to solely or primarily avoid taxes, which are illegal [19].

Finally, governments need access to cross-border tax-related information to be able to administer and enforce tax laws to raise the maximum available resources to fulfil human rights obligations. Most lower-income countries do not have access to the information they need because of legal and administrative barriers designed by the OECD/G20 Global Forum on Transparency and Exchange of Information for Tax Purposes. For instance, only 8 of 54 African countries are currently participating in the exchange of information of country-by-country reports on locally active multinational companies [114]. A differentiated transparency framework may be needed to ensure all countries access information required to assess, query and audit multinational company accounts [69].

## Conclusion

Undermining the rights of citizens across the world is not inevitable. Instead, European governments have a responsibility to change domestic, regional, and international laws and systems that enable multinational companies to artificially shift profits, thereby avoiding

taxes where economic activity occurs. The potential for countries like Nigeria that do not yet have universal coverage of rights is significant. Although individual tax havens seemingly win in the short term due to government revenue gains, globally, profit shifting undermines the international order, and rights for millions of adults and children go unmet.

This paper contributes to the growing body of literature that shows the scale of offshore revenue leakages from lower-income countries has implications for international tax and human rights law [69]. Our study demonstrates that the European extraterritorial impacts of domestic tax policy can be connected to the daily lives of people in other countries. This may help to shift perceptions and engage new groups in tackling global inequalities through international tax reform in a way that is tangible and in reach. This may be through citizens requesting action from elected representatives in national parliaments, such as introducing more robust anti-abuse provisions and stopping the race to the bottom in tax rates that may generate quick wins for government revenue in European countries but to the detriment of non-Europeans from where profits are shifted.

An international order where everyone's economic and social rights are fulfilled will only be possible with international reforms of global tax rules. Reforming how multinational corporations are taxed is foundational for fulfilling human rights and sustainable development; the EU should do more than pay lip service to its founding treaty. Thus, European citizens, who may feel far removed from processes, are encouraged to demand that their governments and the EU support international tax cooperation at the United Nations.

## Author contributions

**Conceptualization:** Rachel Etter-Phoya, Bernadette O'Hare.

**Data curation:** Stuart Murray.

**Formal analysis:** Stephen Hall.

**Funding acquisition:** Bernadette O'Hare.

**Investigation:** Rachel Etter-Phoya, Bernadette O'Hare.

**Methodology:** Stephen Hall, Bernadette O'Hare.

**Project administration:** Bernadette O'Hare.

**Software:** Stuart Murray.

**Supervision:** Stephen Hall, Bernadette O'Hare.

**Visualization:** Stuart Murray.

**Writing – original draft:** Rachel Etter-Phoya, Stuart Murray, Stephen Hall, Michael Masiya, Bernadette O'Hare.

**Writing – review & editing:** Rachel Etter-Phoya, Stuart Murray, Stephen Hall, Michael Masiya, Bernadette O'Hare.

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
