## [Decision Letter · Decision Letter 0]

4 Jul 2024

PGPH-D-24-00819

European tax havens and their impact on fundamental rights in Nigeria and Europe

Dear Dr. Etter-Phoya,

Thank you for submitting your manuscript to PLOS Global Public Health. After careful consideration, we feel that it has merit but does not fully meet PLOS Global Public Health’s publication criteria as it currently stands. Therefore, we invite you to submit a revised version of the manuscript that addresses the points raised during the review process.

We look forward to receiving your revised manuscript.

Kind regards,

Roojin Habibi

Academic Editor

Journal Requirements:

Additional Editor Comments (if provided):

Reviewers' comments:

Reviewer's Responses to Questions

**Comments to the Author**

1. Does this manuscript meet PLOS Global Public Health’s publication criteria ? Is the manuscript technically sound, and do the data support the conclusions? The manuscript must describe methodologically and ethically rigorous research with conclusions that are appropriately drawn based on the data presented.

Reviewer #1: No

Reviewer #2: Yes

2. Has the statistical analysis been performed appropriately and rigorously?

Reviewer #1: I don't know

Reviewer #2: Yes

3. Have the authors made all data underlying the findings in their manuscript fully available (please refer to the Data Availability Statement at the start of the manuscript PDF file)?

Reviewer #1: No

Reviewer #2: Yes

4. Is the manuscript presented in an intelligible fashion and written in standard English?

Reviewer #1: No

Reviewer #2: Yes

5. Review Comments to the Author

Reviewer #1: Thanks for your manuscript entitled

European tax havens and their impact on fundamental rights in Nigeria and Europe.

The study aimed to assess the impact of shifted profits from Nigeria to all tax havens, including European ones, on fundamental rights and governance in Nigeria.

The authors have embarked on a crucial piece of work. The manuscript is not only fascinating but also a significant contribution to the field. The main missing link is the methodology. The authors have critiqued two methods to measure financial flows, but they have not described these methods or the methods they used to generate the results.

Consider revising the title to better reflect the purpose of the manuscript.

The manuscript assesses the impact of illicit financial flows from Nigeria to Europe on rights in Nigeria and Europe.

A comprehensive and clear description of the methodology used will significantly benefit the manuscript. This will provide readers with a better understanding of the research process and the validity of the results. Formula or model specifications and a description of the variables will also be useful.

Indicate the specific period that your research is looking at.

Indicate how you calculated the profits and figured out that the cooperations shifted profits. One would expect international corporations to externalise their earnings, which may not be illegal. Maybe refrain from using the term "all tax havens" unless you know all the tax havens.

Describe what you mean by a tax haven and indicate a list of the leading tax havens for corporations working in Nigeria.

Indicate why the "profit shifting" affected sanitation, clean water and school attendance investments. If you have trend data on the trajectory of the investments in sanitation, water and school enrolment, you may show that "profit shifting" impacted this trajectory.

Minor issues

Please check the manuscript for grammar.

The financial disclosure should include the grant numbers

The data availability should include the specific data you used in the URL.

The total tax revenues in Table 2 do not add up

In table 1, please indicate the source of the information in the table.

The limitations section essentially discusses the research without presenting the limitations. Describe the main limitations of the research, such as the lack of knowledge that if Nigeria did not lose resources from profit shifting, the resources would be invested correctly in sanitation, water, education, etc.

Reviewer #2: This is an important research that examines a fundamental problem being faced by Nigeria in securing economic and social rights for its citizens. The authors have succeeded in offering very innovative arguments and analysis supported by credible data to link the sifting of profits to the loss of capability of the Nigerian state to secure many social determinants of health for their citizens. The aspect dealing with the presenation and analysis of the data was well done. However the bit you struggle is where you make normative arguments and discuss the relevant international human rights instruments and what they mean for the arguments of your paper. In particular, note the following more detailed comments:

• The conflation of Economic and Social Rights (ESR) with fundamental rights on paragraph 47 of the paper is misleading. In the literature, and under international human rights normativity, fundamental rights are associated with civil and political rights for which the principal treaty instrument is the ICCPR. On the other hand, the principal treaty instrument for ESRs is the ICESCR. These categories of rights do not carry the same impetus as civil and political rights. As such they cannot be conflated.

• Para 51 - The extraterritorial enforcement of human rights by states is a questionable proposition given the obligation of state parties to treaties is to safeguard human rights commitments within their territory. You will need to develop your argument here further.

• The link which you make between the increase or decrease in government revenue and human rights protection is a pivotal part of your paper that needs to be more robustly established.

• The table 1 data is interesting. However, is it possible that there is more than one explanation accounting for the poor situation of the underlying determinants of health discussed in your paper that go beyond increase or decrease in revenue?

• Para 80 – the state safeguards rights; not provide rights.

• Para 84 – states are the “duty bearers”.

• The quoted text in paras 143-148 should be indented rather than left intext – it is too long to be an intext quotation.

• Para 162 – You use the acronym IFF which has not been previously used or defined in your paper.

• Consider changing the focus of your paper from fundamental rights to economic and social rights as those are the most appropriate category of rights to which your paper relates. As I mentioned before, fundamental rights are more associated with civil and political rights, a different species of rights.

• Para 325 – check for grammar.

• The normative argument you make on extraterritorial obligations for human rights enforcement which you ground on the UDHR and EU founding treaty opens up a lot space for controversy. Does the UDHR indeed create such an obligation? Which specific articles of the UDHR? The EU treaty is a regional instrument and cannot thus be the basis for the claim in para 358.

• It is true that the UDHR is the basis of support for subsequent international human rights treaties such as the ICCPR and ICESCR, however it will be good to refer your reader to specific articles of the ICESCR where the obligations you mentioned in para 364 of your paper can be found.

• In para 365 you reference General Comment 24. It is not clear which specific treaty or treaty body this General Comment relates to. You need to specify. Also keep in mind that General Comments are soft law and do not create binding statements of law to that extent. However, to the extent that they are interpretive tools for treaties, they carry a weight of their own.

6. PLOS authors have the option to publish the peer review history of their article (what does this mean? ). If published, this will include your full peer review and any attached files.

**Do you want your identity to be public for this peer review?** For information about this choice, including consent withdrawal, please see our Privacy Policy .

Reviewer #1: No

Reviewer #2: **Yes: ** Dr Uchechukwu Ngwaba

---

## [Decision Letter · Decision Letter 1]

8 Jan 2025

Profit shifting from Nigeria to Europe: The impact on human rights

PGPH-D-24-00819R1

Dear Ms Etter-Phoya,

We are pleased to inform you that your manuscript 'Profit shifting from Nigeria to Europe: The impact on human rights' has been provisionally accepted for publication in PLOS Global Public Health.

Best regards,

Roojin Habibi

Academic Editor

Reviewer Comments (if any, and for reference):

Reviewer's Responses to Questions

**Comments to the Author**

1. If the authors have adequately addressed your comments raised in a previous round of review and you feel that this manuscript is now acceptable for publication, you may indicate that here to bypass the “Comments to the Author” section, enter your conflict of interest statement in the “Confidential to Editor” section, and submit your "Accept" recommendation.

Reviewer #2: All comments have been addressed

2. Does this manuscript meet PLOS Global Public Health’s publication criteria ? Is the manuscript technically sound, and do the data support the conclusions? The manuscript must describe methodologically and ethically rigorous research with conclusions that are appropriately drawn based on the data presented.

Reviewer #2: (No Response)

3. Has the statistical analysis been performed appropriately and rigorously?

Reviewer #2: (No Response)

4. Have the authors made all data underlying the findings in their manuscript fully available (please refer to the Data Availability Statement at the start of the manuscript PDF file)?

Reviewer #2: (No Response)

5. Is the manuscript presented in an intelligible fashion and written in standard English?

Reviewer #2: (No Response)

6. Review Comments to the Author

Reviewer #2: (No Response)

7. PLOS authors have the option to publish the peer review history of their article (what does this mean? ). If published, this will include your full peer review and any attached files.

**Do you want your identity to be public for this peer review?** For information about this choice, including consent withdrawal, please see our Privacy Policy .

Reviewer #2: No
